# Proteome-Wide Screening of Potential Vaccine Targets against *Brucella melitensis*

**DOI:** 10.3390/vaccines11020263

**Published:** 2023-01-25

**Authors:** Mahnoor Malik, Saifullah Khan, Asad Ullah, Muhammad Hassan, Mahboob ul Haq, Sajjad Ahmad, Alhanouf I. Al-Harbi, Samira Sanami, Syed Ainul Abideen, Muhammad Irfan, Muhammad Khurram

**Affiliations:** 1Department of Health and Biological Sciences, Abasyn University, Peshawar 25000, Pakistan; 2Institute of Biotechnology and Microbiology, Bacha Khan University, Charsadda 24550, Pakistan; 3Department of Pharmacy, Bacha Khan University, Charsadda 24461, Pakistan; 4Department of Pharmacy, Abasyn University, Peshawar 25000, Pakistan; 5Department of Medical Laboratory, College of Applied Medical Sciences, Taibah University, Yanbu 42353, Saudi Arabia; 6Medical Plants Research Center, Basic Health Sciences Institute, Shahrekord University of Medical Sciences, Shahrekord 8815713471, Iran; 7School of Biomedical Engineering, Shanghai Jiao Tong University, Shanghai 200240, China; 8Department of Oral Biology, College of Dentistry, University of Florida, Gainesville, FL 32611, USA

**Keywords:** *Brucella melitensis*, reverse vaccinology, vaccine, molecular modelling

## Abstract

The ongoing antibiotic-resistance crisis is becoming a global problem affecting public health. Urgent efforts are required to design novel therapeutics against pathogenic bacterial species. *Brucella melitensis* is an etiological agent of brucellosis, which mostly affects sheep and goats but several cases have also been reported in cattle, water buffalo, yaks and dogs. Infected animals also represent the major source of infection for humans. Development of safer and effective vaccines for brucellosis remains a priority to support disease control and eradication in animals and to prevent infection to humans. In this research study, we designed an in-silico multi-epitopes vaccine for *B. melitensis* using computational approaches. The pathogen core proteome was screened for good vaccine candidates using subtractive proteomics, reverse vaccinology and immunoinformatic tools. In total, 10 proteins: catalase; siderophore ABC transporter substrate-binding protein; pyridoxamine 5′-phosphate oxidase; superoxide dismutase; peptidylprolyl isomerase; superoxide dismutase family protein; septation protein A; hypothetical protein; binding-protein-dependent transport systems inner membrane component; and 4-hydroxy-2-oxoheptanedioate aldolase were selected for epitopes prediction. To induce cellular and antibody base immune responses, the vaccine must comprise both B and T-cells epitopes. The epitopes were next screened for antigenicity, allergic nature and water solubility and the probable antigenic, non-allergic, water-soluble and non-toxic nine epitopes were shortlisted for multi-epitopes vaccine construction. The designed vaccine construct comprises 274 amino acid long sequences having a molecular weight of 28.14 kDa and instability index of 27.62. The vaccine construct was further assessed for binding efficacy with immune cell receptors. Docking results revealed that the designed vaccine had good binding potency with selected immune cell receptors. Furthermore, vaccine-MHC-I, vaccine-MHC-II and vaccine-TLR-4 complexes were opted based on a least-binding energy score of −5.48 kcal/mol, 0.64 kcal/mol and −2.69 kcal/mol. Those selected were then energy refined and subjected to simulation studies to understand dynamic movements of the docked complexes. The docking results were further validated through MMPBSA and MMGBSA analyses. The MMPBSA calculated −235.18 kcal/mol, −206.79 kcal/mol, and −215.73 kcal/mol net binding free energy, while MMGBSA estimated −259.48 kcal/mol, −206.79 kcal/mol and −215.73 kcal/mol for TLR-4, MHC-I and MHC-II complexes, respectively. These findings were validated by water-swap and entropy calculations. Overall, the designed vaccine construct can evoke proper immune responses and the construct could be helpful for experimental researchers in formulation of a protective vaccine against the targeted pathogen for both animal and human use.

## 1. Introduction

Antibiotic resistance by bacteria significantly contributes to human morbidity and mortality. This is due to the irrational use of antibiotics in humans, animals, the environment and agricultural fields [1]. Antibiotic resistance is a bacterial evolution process to adjust to changing environmental milieu [2]. Novel approaches are needed to combat this alarming global health concern [3]. The use of bacterial genomic information in vaccine design is a promising approach at the present time. This technique is referred to as reverse vaccinology, which has remarkably changed the field of vaccine design [4]. Vaccines can be used to provoke host immune responses against infectious pathogens. Antigens can be either biological or synthetic in nature [5].

In the beginning of 15th century, smallpox disease resulted in high mortality and morbidity rates. Both Turks and Chinese were trying to induce immunity against smallpox by using powder smallpox lesions [6]. Pasteur’s vaccinology concept was used by Salk and Sabin to design an effective poliovirus vaccine. Reverse vaccinology is a novel tool for vaccine production, which compared to traditional vaccines is a cheap process and can be done in a short time [7]. Identification of new vaccine targets through genomics and computational techniques has not only speed up the vaccine development process but also delivers new antigens not disclosed through experimental methods [8]. For finding putative surface-associated proteins, reverse vaccinology is used without any culturing of microorganisms [9]. Using the same process, meningococcal serogroup B (4DMenB) vaccine was developed [10]. Compared to simple reverse vaccinology, pan-genomic reverse vaccinology is generally more effective as it screens highly conserved targets [11]. As an example, for *Streptococcus agalactiae* four different protective antigens were unveiled by pan-genomic reverse vaccinology [12,13,14,15].

In this study, comparative genomics and reverse vaccinology methods were used for identification of protective vaccine antigens against *Brucella melitensis*, which is a Gram-negative coccobacillus bacterium from the *Brucellaceae* family [16]. *B. melitensis* is an etiological agent of brucellosis, which mostly affects sheep and goats but several cases have also been reported in cattle, water buffalo, yaks and dogs [17]. The pathogen causes brucellosis in goats and Malta fever in humans [17,18]. Infected animals represent the major source of infection for humans, through direct exposure or through consumption of contaminated and unpasteurized dairy products. In endemic areas, vaccination of susceptible animals would reduce disease prevalence also limiting the risk of disease transmission to humans [19]. Previous efforts on *B. melitensis* vaccination are dominated by work on the Rev1 vaccine which is effective against sheep and goat brucellosis [20]. The vaccine comprises smooth lipopolysaccharide with O-polysaccharide able to elicit strong antibody responses [21]. Rev1 is a live attenuated vaccine and despite its efficacy, several drawbacks remain due to its residual pathogenicity. The Rev1 vaccine may also result in abortion in pregnant animals [22,23]. Previous attempts to produce safer and effective recombinant brucella vaccines included two antigens (periplasmic bp26 and chaperone trigger factor proteins); however, both candidates were not able to induce protective and accurate immune responses [22,24]. Despite these efforts, no licensed vaccine is available to prevent brucellosis infection. In addition, drug-resistant strains of *B. melitensis* are making the situation worse. Thus, considering this, herein we applied an integrated approach comprising comparative genomics, subtractive proteomics, reverse vaccinology, immunoinformatic, and biophysics techniques to identify protective antigens from *B. melitensis* completely sequenced genomes and designed a multi-epitopes vaccine [25,26,27,28,29,30,31,32,33]. The designed vaccine construct then was examined for interactions with host immune receptors to check whether the vaccine is able to be presented to the host immunity system. The findings of this study may help in formulating vaccines design against *B. melitensis*.

## 2. Materials and Methods

The study framework used for designing a multi-epitopes vaccine against *B. melitensis* is illustrated in Figure 1.

### 2.1. Proteomes Retrieval of B. melitensis

The complete proteomic data of fully sequenced *B. melitensis* strains were extracted from the national center for biotechnological information (NCBI) genome database [34]. The proteomic data were retrieved in FASTA format.

### 2.2. BPGA Analysis

For identification of suitable vaccine candidates, all the proteomes were screened with the bacterial pan genomic analysis (BPGA) tool [35]. During this analysis, different sets of proteins were identified such as core, accessory and unique. The core proteins were picked and used in downward analysis as they have broad spectrum applicability [36,37,38,39].

### 2.3. CD-Hit Analysis

In the pre-screening phase, the core sequences were subjected to redundancy checks to remove duplicate copies of the proteins [40]. The non-redundant proteins are single presentations in the proteome and do not require extra computational cost. In the CD-hit (cluster data with high identity with tolerance) analysis, core proteins were analyzed for the presence of redundant proteins [41,42,43]. Non-redundant proteins were selected as the best candidates for the vaccine design [44,45]. The core proteins were clustered at a threshold of 0.5 which means that the sequences that were 50% similar were clustered together.

### 2.4. Subcellular Localization Phase

In the field of vaccine design, surface proteins elicit robust immune response [46,47,48]. The non-redundant proteome was examined though PSORTb 3.0, which is a bacterial protein subcellular localization prediction program [49].

### 2.5. Homology Check

In the homology check, the shortlisted proteins were blast through the BLASTp (basic local alignment search tool) against the human proteomes for sequence similarity [50]. Those proteins with E values < 1.0^−4^, bit scores <100 and <30% sequence identity were selected [36,37]. Homologous proteins provoke autoimmune reactions [51]. A similar homology check was also performed against intestinal probiotic bacteria *Lactobacillus rhamnosus* (taxid: 47715), *L. casei* (taxid: 1582) and *L. johnsonii* (taxid: 33959) to avoid their function inhibition [52].

### 2.6. Vaccine Candidate’s Prioritization Phase

In the vaccine candidate prioritization phase, the main focus was on prioritization of potential vaccine candidates for vaccine design.

### 2.7. Virulent Protein Analysis

Virulent proteins are good vaccine candidates as they have the ability to stimulate immune responses of the host [53]. The proteins were blast using BLASTp against the core virulent factor database (VFDB) [54]. The proteins selected as the best choices for vaccine design exhibited a sequence identity ≥30% and a bit score >100. Those proteins that were below the set parameters were discarded [55].

### 2.8. Physiochemical Analysis

The filtered proteins were analyzed for physiochemical properties including, molecular weight, atomic composition, instability index, theoretical PI, amino acid composition, aliphatic index, grand average of hydropath city (GRAVY) and estimated half-life [56]. These properties were analyzed using the ProtParam-2017 tool [57]. Those proteins having a molecular weight less than 110 kDa and thermostability index less than 40 were considered to be good candidates for vaccine design as they can be easily purified [58].

### 2.9. Transmembrane Helices

The proteins that have less transmembrane helicase were considered as good candidates for vaccine design [31,33]. For the analysis of transmembrane helicase, HMMTOP 2.0 [59] and TMHMM 2.0 [60] softwares were used. The threshold values were set at 0 and 1. Proteins that were exposed and had values of 0 and 1 were selected [32].

### 2.10. Antigenicity Prediction

For stimulation of the host immune system, foreign antigens should be capable of binding with the host immune cells; that capability of antigens is known as antigenicity [61]. For the detection of antigenicity of shortlisted proteins, the online server VaxiJen 2.0 was used [62]. The threshold value for the antigenicity of proteins was set at 0.4. Those proteins that had antigenicity scores >0.5 were considered as good choices for vaccine design [47].

### 2.11. Adhesion Probability Analysis

For stimulation of the host immune system, vaccine antigens should be able to attach to the host cells and start the infection process to be recognized as antigens by the host immune system [55]. After attachment, adaptive immunity is developed which includes T cell receptors and antibodies [63]. For the analysis of the adhesion probability of proteins, the online server Vaxign was used [64]. The threshold for the selection of a good protein was set at >0.5 [36].

### 2.12. Allergenicity of the Proteins

Those proteins that were allergens to the host were removed and only non-allergenic proteins were selected. The allergenicity of the proteins was determined through the online webserver AllerTOP 2.0 [65].

### 2.13. Epitopes Prediction 

The epitope is the main part of the antigen to which the host immune cells bind. In prediction of epitopes, the B cell and T cell epitopes were analyzed via the immune epitope database (IEDB) tool [66]. The threshold for the IEDB server was set at 0.5. In the epitope-prediction phase, the B cell epitopes were first predicted and then T cell epitopes. The T cell epitopes were analyzed for the prediction of potential binding alleles for the major histocompatibility complex (MHC) class I and MHC class II. Prediction of binding of epitopes with the DRB*10101 alleles was analyzed using the MHcPred tool [67]. The threshold value set was IC50 values ≤100 nM [68]. Those exceeding this threshold value were discarded.

### 2.14. Physiochemical Analysis of the Predicated Epitopes

The selected epitopes were then subjected to physiochemical properties analysis. The antigenicity and allergenicity of the epitopes were analyzed using VaxiJen 2.0 [62] and AllerTop 2.0 [65], respectively. The virulence property was checked via Virulentpred [69]. The antigenic and non-allergen epitopes were further analyzed using ToxinPred for toxicity [70]. All those epitopes having good water solubility were considered as good candidates. The ProteinSol tool (https://protein-sol.manchester.ac.uk/; accessed on 16 October 2022) was used for the prediction of water solubility.

### 2.15. Multi-Epitopes Peptide Designing

Peptide vaccines have weak ability to provoke host immune responses. The weak immunogenicity of the peptide vaccine can be handled with the technique of multi-epitopes peptide design [71,72]. In multi-epitopes peptide-design phase, immunodominant epitopes were linked to each other [73]. All the selected screened epitopes were linked with the Gly-Pro-Gly-Pro-Gly (GPGPG) linkers to build an immunopotent multi-epitopes peptide vaccine. Furthermore, the vaccine was linked with the highly immunopotent cholera toxin B adjuvant (CTB) for the enhancement of the vaccine immunogenicity [74,75]. The adjuvant binds to the monoganglioside GM1 receptor and is capable of stimulating cytokines, interferone, cellular and humoral immunity [75]. The adjuvant is used in vaccine design against cancer, tuberculosis and influenza [76].

### 2.16. Physiochemical Properties Analysis

The final vaccine construct was then subjected to various physiochemical properties. The online server ExPASy Protparam was used for the evaluation of molecular weight, number of amino acids, instability index, Grand average of hydropathicity (GRAVY), theoretical PI value and aliphatic index [57].

### 2.17. Structural Prediction of Multi-Epitope Peptide

The final vaccine construct was then run on the 3Dpro tool of the SCRATCH protein predictor for construction of the tertiary structure [77]. A good vaccine consists of a smaller number of loops, is small in size and simple in structure. For the removal of loops, the final vaccine construct was then run on the Galaxy Loop tool of the GalaxyWeb online server [78].

### 2.18. Galaxy Refinement

For the final refinement of the loop-modelled vaccine construct, the GalaxyRefine tool was used for the construction of side chains and removal of steric clashes [79]. The fully refined form of the vaccine construct was considered to be good candidate for vaccine design.

### 2.19. Disulfide Engineering

The stability of a vaccine is very important; thus, for better stability the bonding of the inner and outer chains was checked in silico. At the disulfide engineering phase, the Design 2.0 webserver was used for the introduction of disulfide bonds in the vaccine construct [80].

### 2.20. In Silico Codon Optimization and Coding

The sequence of the multi-epitope vaccine construct was translated into a DNA sequence and then cloned into the expression vector to be expressed in *Escherichia coli*. The conversion of the vaccine into DNA was performed via the Java Codon Adaptation Tool (JCat) [81]. The expression level of the cloned sequence was evaluated through the “GC’ concentration and the “Codon Adaptation Index score (CAI)”. Preferably, the CAI score should be 1 and the GC content should be 30–70% [82].

### 2.21. Docking and Refinement

The binding affinity of the multi-epitope vaccine construct with the immune cell receptors was analyzed through blind molecular docking [83]. In the PatchDock server [84], the TLR-4, MHC-I and MHC-Ⅱ receptors were selected. The tertiary structures of these receptors were collected from the Protein Data Bank (PDB) by using the codes 4G8A, 1I1Y and 1KG0, respectively. The output solutions were generated by the PatchDock server and were then subjected to further refinement via the online webserver fast interaction refinement in molecular docking (FireDock) [85]. After the refinements, those complexes that exhibited the least global energy were ranked at the top and were selected for further analysis. The intermolecular interactions of complexes were analyzed via UCSF Chimera software [86].

### 2.22. Molecular Dynamics Stimulation (MDS) Assay 

The dynamic behavior of the vaccine complexes was analyzed by using a molecular dynamics stimulation approach. Based on the lowest global energy value, the complexes were chosen for the MDS [87]. The AMBER20 stimulation software was used for simulation on a timescale of 200 ns [88]. For the completion of system setup phase, preprocessing and final production phase, the AMBER SANDER module was used [89]. The intermolecular interactions in the MDS were defined by FF14SB force field [90]. SHAKE algorithm was used to constrain hydrogen bonds [91]. The pressure equilibrium of the system was maintained by NPT ensemble. For the evaluation of trajectories, the CPPTRAJ module was used [92]. To investigate structure stability of complexes, root mean square deviation (RMSD) and root mean square fluctuation (RMSF) plots were produced in XMGRCE [93].

### 2.23. Free Energy of Immune Receptors and Vaccine Design

The tool MMPBSA.py provided in AMBER20 was used for the determination of binding free energies of the docked complexes [94,95,96]. In binding free energy analysis, a total of 100 frames were selected from the trajectories. Vaccine immune receptor complexes are stable having lower free binding energies [48].

### 2.24. WaterSwap Validation and Entropy Analysis

The MMPBSA often ignore the water molecule contribution in bridging the vaccine and receptors residues. Therefore, water-swap analysis was conducted with default settings to reconfirm stable intermolecular interactions [97]. Also, entropy energy calculations were done on 5 frames using AMBER normal mode analysis [98].

## 3. Results

### 3.1. Retrieval of Complete Proteome, Bacterial Pan-Genome Analysis and Subtractive Proteomics Filters

The study was commenced with the retrieval of 95 fully sequenced strains of *B. melitensis*. The extracted strains were subjected to bacterial pa-genome analysis steps for retrieval of core sequences. In total, 238,450 core sequences were predicted. The core proteins were good targets for broad spectrum vaccine development. The core sequences were further considered for redundancy analysis. The webserver revealed that the 238,450 core sequences consisted of 2551 non-redundant proteins and 235,899 redundant proteins. The redundant proteins were discarded, and the non-redundant proteins were further considered for surface-localization analysis. The non-redundant proteins are single presentations in the proteomes and require less computational expense to process them [40]. The analysis revealed that non-redundant proteins consisted of 26 outer-membrane, 9 extracellular and 71 periplasmic membrane proteins. Next, VFDB analysis was performed, which predicted that the surface localized proteins consisted of 12 virulent proteins. Virulent proteins are good vaccine targets as they have antigenic epitopes capable of stimulating immune responses [46]. In 12 virulent proteins, 3 proteins were predicted to have more than 1 transmembrane helix. Overall, 12 proteins were predicted to be probably antigenic, have good water solubility and be non-toxic. Moreover, no physiochemically unstable, host or normal flora similar proteins were found. The category and number of proteins are presented in Figure 2. The size of each proteome is presented in Figure 3.

### 3.2. Epitopes Prediction 

B and B-cell derived T-cell epitopes were predicted from the shortlisted 11 proteins given in Table 1. The predicted B-cell epitopes were further utilized for T-cell epitopes. The predicted MHC-I and MHC-II epitopes are tabulated in Appendix A with least percentile scores. The common and lowest percentile score epitopes were opted for downward analyses.

### 3.3. Selection of Epitopes for Multi-Epitopes Vaccine Construction

Final set epitopes were selected for vaccine construct by applying several filters. Only antigenic, allergenic, water solubility and non-toxic B and B-cell derived T-cell epitopes were considered. Table 2 shows selected epitopes utilized in multi-epitopes vaccine construction. The schematic diagram of vaccine construct comprising selected epitopes is presented in Figure 4.

### 3.4. Structure Prediction and Disulfide Engineering

The vaccine 3D structure was predicted using a scratch predictor. The structure of the vaccine comprises cholera toxin B subunit as an adjuvant molecule, EAAAK and GPGPG linkers along with selected epitopes as presented in Figure 5. The EAAAK and GPGPG linkers are rigid in nature and keep the epitopes separated which will allow the epitopes to be recognized by the host immune system for efficient recognition and processing [38]. The adjuvant cholera toxin B subunit is considered a powerful adjuvant as it generates specific immunity and mucosal antibody responses. The adjuvant binds to GM1 ganglioside, which is present on antigen-presenting cells, lymphocytes and epithelia cells. The adjuvant conjugation to antigens can result in activation of dendritic cells, decrease in antigen dose, and enhanced B and T-cell responses [75,76,99]. Furthermore, to retain structural stability of the vaccine, disulfide bonds were established between weak energy pairs. The enzymatic sensitive residue bonds were supplemented by cysteine bonds. The pairs of amino-acid residues that were disulfide engineered are tabulated in Table 3 and presented in Figure 6 by yellow sticks.

### 3.5. Loops Refinement

In the loop-refinement phase, the structure of the designed vaccine construct was examined for loops refinement. A total of 10 refine models were generated and model 1 was selected for docking analysis. The top 10 refined models are tabulated in Table 4. The refined model has a good MolProbity score of 1.475, improved clash score of 2.3 and high percentage of residues in Rama-favored regions (92.6).

### 3.6. Codon Optimization Phase

The sequence of the multi-epitopes vaccine construct was translated into a DNA sequence and then cloned into the expression system of *Escherichia coli*. This was carried out to get higher expression of the cloned vaccine as E. coli is a good expression system. The CAI score of the vaccine was 0.95 and the GC content was 49.56%. Both these values indicate good expression of the vaccine construct might be expected. Cloning of the vaccine construct is mentioned in Figure 7.

### 3.7. Docking and Refinement

Molecular docking analysis was carried out in order to analyze the binding mode of the vaccine construct with immune cell receptors (MHC-I, MHC-II and TLR-4). The patch dock web server generated the top 20 docked complexes as tabulated in Appendix A. The docked complexes were further refined using the fire docked webserver. The server generated the top 10 refined docked complexes as mentioned in Table 5, Table 6 and Table 7. In the case of the vaccine-MHC-I complex, solution 1 was selected as it has the lowest global energy score of −5.48 kcal/mol. The major energy contribution was seen from attractive van der Waals energy. For vaccine-MHC-II, solution 9 was opted as it has lowest global energy of 0.64 kcal/mol. Similarly, for the vaccine-TLR-4 complex, solution 7 was opted for with a net global energy score of −2.69 kcal/mol. In all the selected complexes, it was found that the vaccine docked with receptors in a stable conformation and the vaccine antigens were exposed to and recognized by the host immune system. The intermolecular docked complexes are presented in Figure 8.

### 3.8. MDS Analysis

MDS analysis was carried out to check the binding stability and dynamics of docked complexes. The docked complexes were investigated at 200 ns periods of time. The trajectories of the MDS consisted of RMSD and RMSF analyses [100,101]. As compared to vaccine-MHC-II and TLR-4, vaccine and MHC-I molecules showed stability as the graph is constant through the simulation time. The vaccine-MHC-II complex showed minor changes which may be due to loops present in the structure but towards the simulation end it became stable. Overall, the RMSD graph shows that there is proper conformational stability between vaccine and receptors molecules. The RMSD graph is presented in Figure 9A. Similarly, residue base fluctuations were assessed through RMSF. In RMSF analysis, it was observed that the vaccine-MHC-II complex has proper stability as mentioned in Figure 9B.

### 3.9. Binding Free Energies Calculation

The docking results were further validated using binding free energies calculations. The MMGB/PB/SA analysis was adopted for energy estimation. In MMGBSA analysis, −259.48 kcal/mol, −206.79 kcal/mol, −215.73 kcal/mol delta energy were estimated for vaccine-TLR-4, vaccine-MHC-I and vaccine-MHC-II, respectively. Similarly, in MM-PBSA analysis, for vaccine-TLR-4, vaccine-MHC-I, and vaccine-MHC-II, a net energy of −235.18 kcal/mol, −206.79 kcal/mol and −215.73 kcal/mol was calculated. Details of overall binding energies prediction are tabulated in Table 8.

### 3.10. WaterSwap and Binding Entropy Calculation

To revalidate the findings and provide more confidence regarding the vaccine’s stable interactions with the receptors, water-swap calculations were performed. Three algorithms were used in the water-swap method i.e. thermodynamic integration, free energy perturbation and Bennetts. The water-swap calculations found highly stable energies for all three complexes [97,102,103]. For the TLR-4-vaccine complex, the water-swap estimations were: thermodynamic integration (−46.5 kcal/mol); free energy perturbation (−47.52 kcal/mol); and Bennetts (−46.97 kcal/mol). The thermodynamic integration, free energy perturbation and Bennetts values for the MHC-I-vaccine complex were −48.5 kcal/mol, −47.0 kcal/mol and −47.41 kcal/mol, respectively. Likewise, for the MHC-II-vaccine complex, the values were −42.1 kcal/mol (thermodynamic integration), −43.6 kcal/mol (free energy perturbation) and −41.6 kcal/mol (Bennetts). All these values indicate good systems convergence and strong intermolecular affinity. Additionally, entropy energy indicates values of 45 kcal/mol for the TLR-4-vaccine complex, 51.87 kcal/mol for the MHC-I-vaccine complex, and 50.2 kcal/mol for the MHC-II-vaccine complex. These findings suggest that the vaccine has less physical freedom and docked well with the receptor for efficient immuen system recognition and processing.

## 4. Discussion

In this study, nine proteins were prioritized as potential subunit vaccine candidate targets in the complete proteome of *B. melitensis* based on comprehensive investigation of comparative proteomics, subtractive proteomics, reverse vaccinology, immunoinformatic, and biophysics approaches. Furthermore, antigenic, non-allergic, non-toxic, and water-soluble epitopes were successfully predicted in the mentioned vaccine proteins. A multi-epitopes vaccine construct was built which revealed stable binding conformation and dynamics with different immune receptors such as TLR-4, MHC-I and MHC-II.

The emergence of bacterial resistance to antibiotics is a serious threat to public health [104]. The resistance to antibiotics is alarming as efficacy of commercially available antibiotics is becoming less effective. Additionally, development of new antibiotics is of less interest due to several regulatory challenges faced by pharma companies [105]. This problem can be addressed by a vaccination process [15]. However not all vaccines are effective and helpful in prevention of infectious disease but somehow can reduce the level of infections [106,107]. The conventional Pasture vaccinology suffers from several limitations i.e., time consuming, development of unwanted immune responses, high cost, less efficacy and specificity, not applicable to non-cultivated microbes and less stability [108]. A huge amount of genomic data available in databases could help in identification of good vaccine candidates [12,13,14]. The use of reverse vaccinology, bioinformatics and immmunoinformatics approaches in recent times is an alternative way of designing vaccines against different pathogens [11,15,109]. We used pan-genome analysis, subtractive proteomics analysis immunoinformatics, and molecular docking and simulation methods for the designing of multi-epitopes-based vaccines against *B. melitensis.* The multi-epitopes vaccines consist of overlapping epitopes and are considered an ideal approach for prevention and treatment of infectious diseases [107]. The success of these vaccines has been elaborated by the EMD640744 vaccine that is currently under phase I clinical trials for advanced solid tumors [107]. These vaccines can generate humoral, cytotoxic and helper T-cell immune responses. They contain epitopes which can be recognized by multiple clones of TCRs and stimulate cellular and humoral immunity simultaneously [107]. They also have the ability to generate enhanced immunogenicity that is long term [107]. Additionally, they are free from unwanted antigens that lead to pathologically adverse reactions [71]. Through subtractive proteomics filters, 10 proteins were selected for B–cell peptides prediction. These proteins were catalase, siderophore ABC transporter substrate-binding protein, pyridoxamine 5′-phosphate oxidase, superoxide dismutase, peptidylprolyl isomerase, superoxide dismutase family protein, septation protein A, hypothetical protein, binding-protein-dependent transport systems inner membrane component, 4-hydroxy-2-oxoheptanedioate aldolase. The shortlisted proteins were utilized for B-cell and T-cell epitopes in the prediction and prioritization phases. With the help of several immunoinformatics approaches, WTNAEAEQV, EAEAKAEAE, EADAYYASR, GLEGKSLEE, KGFDAARVG, YAPEPQPQT, REISAAEGR, KSGVSGNRL, and ESDQTGSSP were prioritized as appropriate epitopes for multi-epitopes vaccine design. The epitopes were tested for antigenicity, allergenicity, toxicity, water solubility, and adhesion probability. Further, non-toxic, probable antigenic, good water-soluble and physiochemically stable epitopes were used in chimeric vaccine construction. The designed vaccine was then utilized for interaction studies with immune cell receptors as it was important to unveil for successful vaccine development. This was achieved by molecular docking that predicted the vaccine candidates’ stable interactions with MHC-I, MHC-II and TLR-4 and thus could evoke humoral and cellular immunity. Intermolecular docked stability of vaccine with immune cell receptors in dynamic environments is important for long-term antigen presenting and processing. The dynamic movement of the docked complexes revealed the vaccine candidates’ proper binding that can generate long-term immunity against the targeted pathogen.

Brucellosis is estimated to cause 500,000 thousand cases each year. Among the Brucella genera, *B. melitensis* is the most pathogenic species and shows broad resistance to a spectrum of antibiotics especially rifampicin [110]. Therefore, efforts are needed for development of a safe and effective vaccine. In the recent past, several computational efforts have revealed potential vaccine candidates against different bacterial pathogens. A previous study conducted by Ismail, Ahmad and Azam, 2020, successfully predicted an in silico multi-epitopes vaccine against bacterial members of Enterobacteriaceae [38]. In another work, three distinct types of surface peptides were investigated that can effectively provoke the immune response (AtfC), (PMI2533) and (PMI1466) against *Proteus mirabilis* [111]. Furthermore, computer-aided vaccine-design studies against *Pseudomonas aeruginosa* [46], *Providencia rettgeri* [112], *Streptococcus pneumoniae* [113], and *Klebsiella pneumoniae* [114] have been successfully carried out in the recent past. The reverse vaccinology approach have also been applied to *B. melitensis*. However, those studies were not as comprehensive as that conducted herein. In one study, Omp10, Omp25, Omp31 and BtpB were used for epitopes prediction. The designed multi-epitopes vaccine comprised 806 amino acids that were used in different biophysics approaches [115]. In another work, Omp22, Omp28 and Omp19 were utilized for vaccine design [116].

In silico vaccine design is rapidly emerging due to the wide range of genomic data available and development of new bioinformatics tools. These analyses are highly effective and provide new pathways for the synthesis of novel vaccines against resistant pathogens [9,117,118,119]. However, our study has a few limitations. The order of the epitopes in the designed vaccine must be evaluated for optimal biological potency [38]. The choice of delivery route and delivery system are also a challenge [38]. Detailed experimental testing is required to validate the immune potency of the designed vaccine against *B. melitensis*.

## 5. Conclusions

In this computer-aided vaccine-design work, 10 proteins; catalase, siderophore ABC transporter substrate-binding protein, pyridoxamine 5′-phosphate oxidase, superoxide dismutase, peptidylprolyl isomerase, superoxide dismutase family protein, septation protein A, hypothetical protein, binding-protein-dependent transport systems inner membrane component and 4-hydroxy-2-oxoheptanedioate aldolase were identified as promising vaccine targets selected for epitopes prediction against *B. melitensis.* The proteins were forecasted to harbor antigenic epitopes that were capable of eliciting strong humoral, cellular and helper immunological responses. Most of the mentioned targets were not predicted before and are novel in this respect. The designed chimeric multi-epitopes vaccine showed a robust interactions network with different immune receptors ensuring that the vaccine is capable of eliciting a variety of immunological reactions. The vaccine type is safe from allergic and reactogenic responses and accurate/specific in generating immunological response and memory. Further, the multi-epitopes vaccine will be easy to design and could provide better immunogenicity and antigenicity compared to single peptide vaccines. The vaccine construct may also provide experimentalists a ready framework to test vaccine epitopes in vivo and in vitro biological models and thus may speed up the development of a safe, effective and broad spectrum vaccine against *B. melitensis*.

## Figures and Tables

**Figure 1 vaccines-11-00263-f001:**
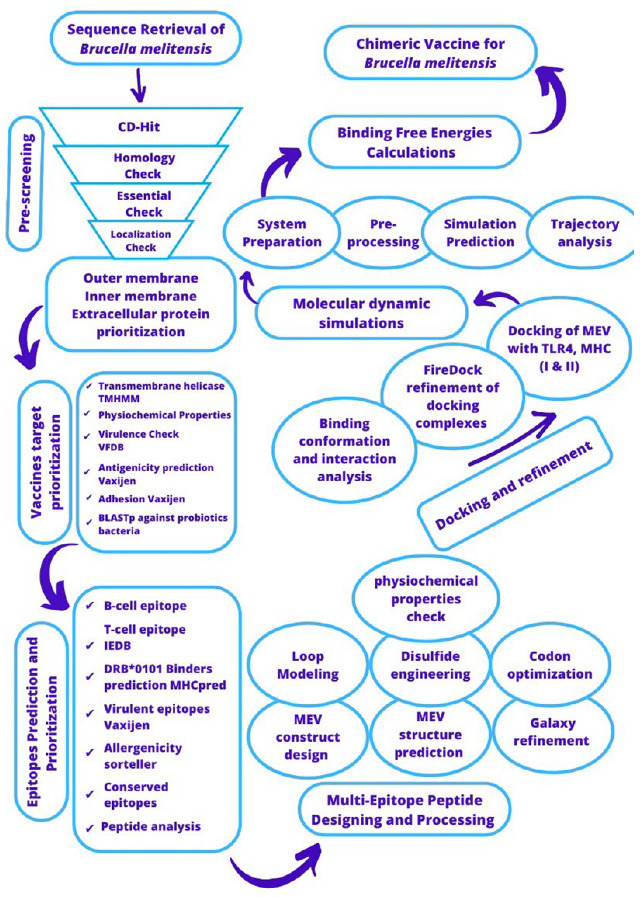
Outline of methodology that was used for designing a multi-epitopes vaccine against *B. melitensis*.

**Figure 2 vaccines-11-00263-f002:**
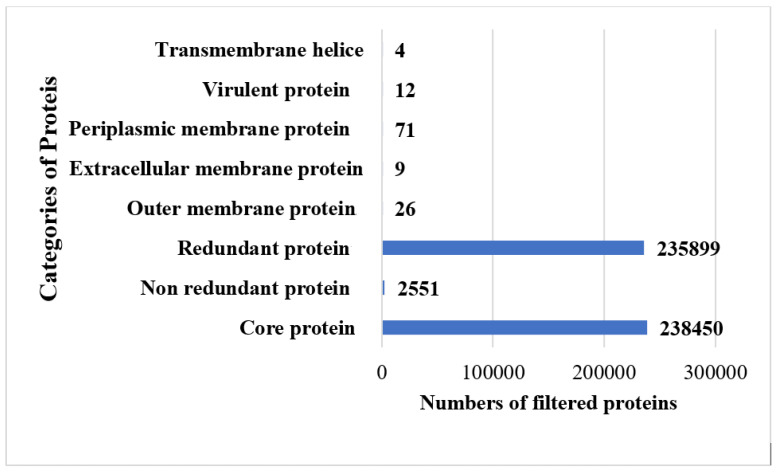
Shortlisted proteins in each step of subtractive proteomics filter.

**Figure 3 vaccines-11-00263-f003:**
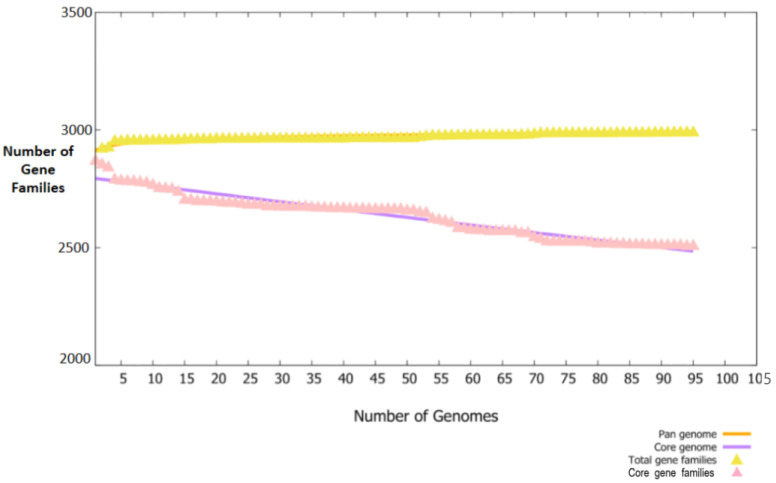
Pan-core plot. The number of fully sequenced proteomes used are plotted on the X-axis while gene families are plotted against each proteome on the Y-axis.

**Figure 4 vaccines-11-00263-f004:**
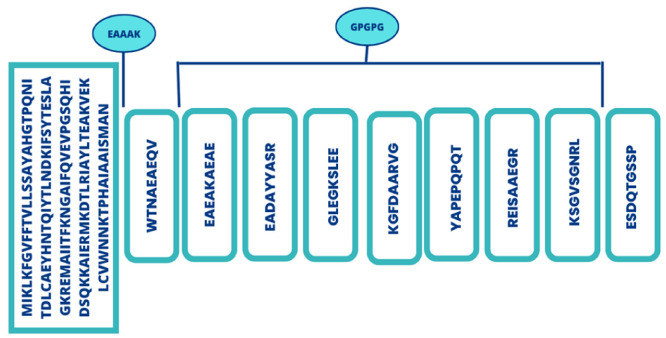
Schematic representation of multi-epitopes vaccine.

**Figure 5 vaccines-11-00263-f005:**
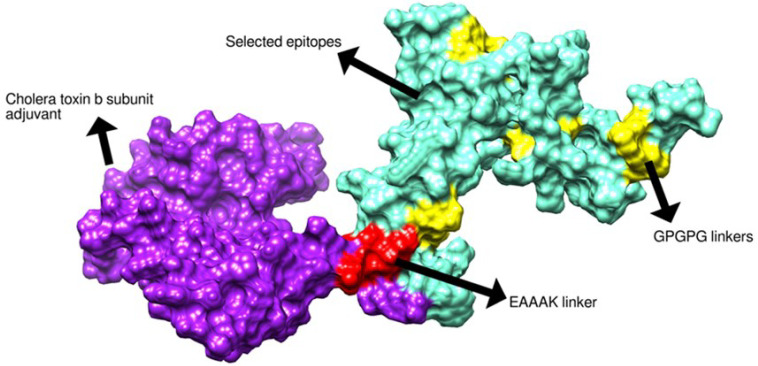
Three-dimensional structure of designed vaccine.

**Figure 6 vaccines-11-00263-f006:**
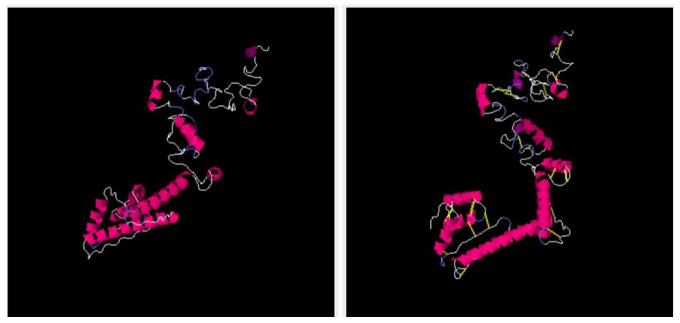
Left side picture represents wild type structure while right side picture represents mutated structure.

**Figure 7 vaccines-11-00263-f007:**
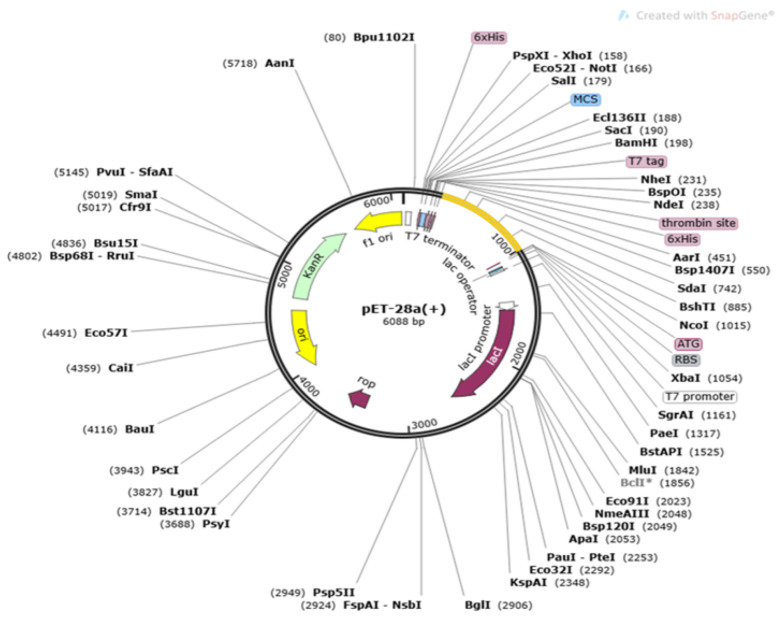
Cloning of multi-epitopes vaccine constructs (yellow color) into pET28a (+) vector.

**Figure 8 vaccines-11-00263-f008:**
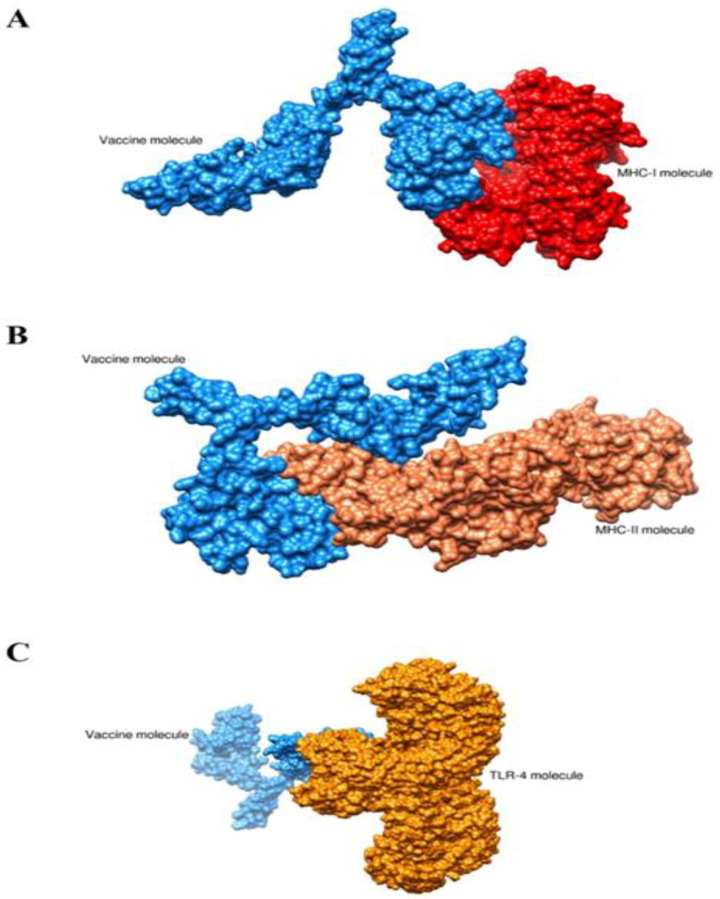
Intermolecular docked conformation of complexes. (**A**) vaccine-MHC-I complex, (**B**) vaccine-MHC-II complex and (**C**) vaccine-TLR-4 complex.

**Figure 9 vaccines-11-00263-f009:**
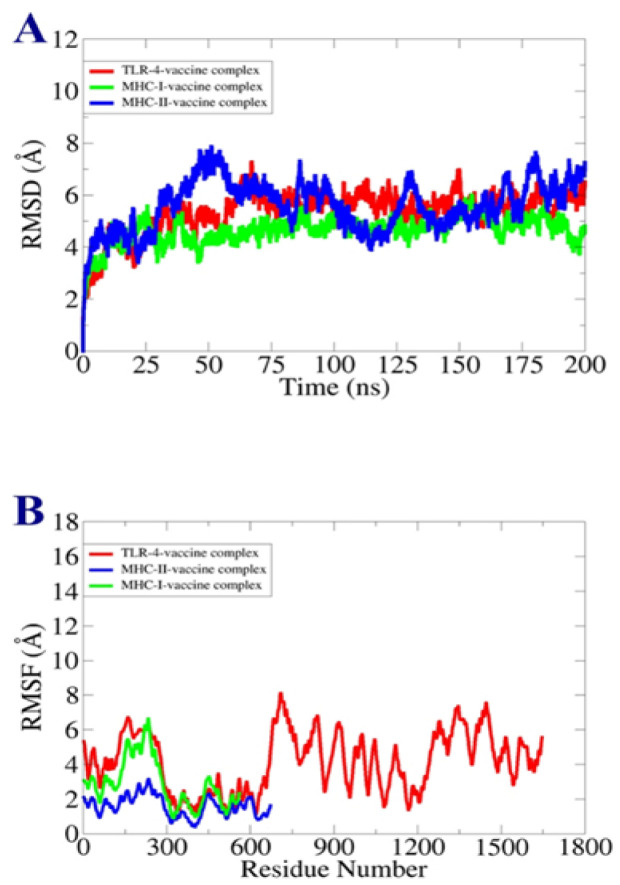
Simulation trajectories (**A**) RMSD (**B**) RMSF.

**Table 1 vaccines-11-00263-t001:** Predicted B-cells epitopes. The epitopes vary in length and have higher scores than the threshold.

Proteins	Predicted B-Cells Epitopes
core/352/1/Org1_Gene809 (catalase)	GAPIPDNQNSLTAGERGPILMQ
	KRHPRTHLRSAT
	QGHKHWTNAEAEQVIGRTREST
	HRLGTHYESIPVNQPKCPVHHYH
	GIKTGNPDAYYEPNSFNGPVEQPSAKEPPLCISG
	ADRYNHRIGNDDYS
	LKDAHGYDANTIALNEKI
	KRHPRTHLRSAT
	QGHKHWTNAEAEQVIGRTREST
core/1230/1/Org1_Gene1297 (siderophore ABC transporter substrate-binding protein)	VPFPEYLKKYQGDDYAKVGTLFEPDYEAVNA
	EAEAKAEAEKLNKELAA
	PAAPNLSIGNHGQPISSE
	DAAIGREGNSAKQ
core/2014/1/Org1_Gene1274 (pyridoxamine 5′-phosphate oxidase)	SSDDFTQSAEPF
	DAEADAYYASRPR
	QSRPLESRFALE
core/2047/1/Org1_Gene1622 (superoxide dismutase)	LPALPYDYDALAPFMSRE
	GLEGKSLEEIVK
core/2062/1/Org1_Gene1723 (peptidylprolyl isomerase)	VKFGNMDKGFDAARVGTGGSNYPDLPAEFSKEPF
core/2194/1/Org1_Gene1299 (superoxide dismutase family protein)	SCAPGEKDGKIVPA
	HYDPGNTHHHLGPEGDGHMG
	GDNYSDKPEPLGGGG
core/123/4/Org4_Gene2756 (septation protein A)	DGSKKTPEQLDRERRLAQAMID
	LAGNIKQARADNAEKAGIEGAAKKFAGLDLGSLLSGGAAYPSAVAGGASPTSGAATGTTPTTGATVDLSGDKQKF
	VINGQRVKINDSFRTFASP
	INLPQQAQPQGVQVASLDPSIGMAQAYAPEPQPQTAAAAINQIAPQQPVPEAKISDALLRQNDMALGGALAPQGQAPQQVADTSGYFPAAPSADSAPIMGSYAAPRQGGVN
	DALRAKPQTEYGFTTLPDGTVLRTDKRSGNAEPIYSAGQKPTSDMQEY
	FAVSQGFKGSFADYQQAMKKAGASSTNVSVGEGDKFYEALDKKNAD
	DAGIQARSKLAQIERLGGLMQASPT
	LVPQQRQPGSGPMSDA
	QYQIQMGDIADQVANREISAAEGRNRIKNLKNPLEGFRTSTKDKTPGKSGVSGNRLRFNPQTG
core/389/1/Org1_Gene1317 (hypothetical protein)	SVVSRNISGAKDADYSRR
	SALYSADNYSGSPSG
	VVGGTRMGRDVSDYLDQRDAL
	ARKVTFEQSAVLTPGVAGKAVTVDGVPLSHDTFDQPFGTG
	ESDQTGSSPDQTGLFSWSGSPAIPGAGLSAGIAGTIEVSVPFIASEGGSALLLRDGGANGANYKYNVQGAAGFSDRLRALNEAFSEPMVFDAAAGISSSSSLIGYS
	KRQKANSEFTYNGT
	FALSNATGVDID
core/2225/1/Org1_Gene971 (Binding-protein-dependent transport systems inner membrane component)	SKKNLPNNAGDLGLGAGAATPGSSQ
	ISYGNERPVAVCDADTCWSQ
core/2432/1/Org1_Gene927 (4-hydroxy-2-oxoheptanedioate aldolase)	SPVGSNTTNSASTASNSTSAANKASVDYD
	NQDPTQPMDPTQY

**Table 2 vaccines-11-00263-t002:** Selected epitopes for vaccine construct.

Selected Epitopes	Antigenicity	Allergen City	Water Solubility	Toxicity
WTNAEAEQV	7.637	Non-Allergenic	Good water soluble	Non-toxigenic
EAEAKAEAE	9.62			
EADAYYASR	0.93			
GLEGKSLEE	1.9			
KGFDAARVG	1.7			
YAPEPQPQT	1.8			
REISAAEGR	1.7			
KSGVSGNRL	0.9526			
ESDQTGSSP	2.0515			

**Table 3 vaccines-11-00263-t003:** Pairs of amino-acid residues Chi-3 values and energy.

Amino Acid Residues Pairs	Chi3	Energy
Ser 15-Pro23	72.16	2.8
Tyr 18-Thr22	−69.94	3.3
His 20-Ala59	124.08	4.12
Gln 24-Glu57	112.52	2.75
Ile 38-Leu41	110.17	4.3
Thr 49-Ala53	102.94	6.53
Phe 69-His78	−73.43	2.77
Gln 70-Val73	105.4	1.08
Ala 101-Ala123	124.39	4.67
Cys 107-Lys112	112.51	0.94
Gly 135-Asn142	102.4	6.24
Asn 161-Ala183	110.79	4.33
Lys 196-Ser201	110.02	1.36
Pro 222-Gln229	109.88	1.95
Pro 250-Ser253	87.28	5.32
Ser 253-Gly257	97.61	2.49
Gly 265-Ser272	−101.99	1.73

**Table 4 vaccines-11-00263-t004:** Model, RMSD, MolProbity, clash score, poor rotamers, Rama-favored and GALAXY energy of refined complexes.

Model	RMSD	MolProbity	Clash Score	Poor Rotamers	Rama Favored	Galaxy Energy
Initial	0.000	3.643	92.4	6.0	87.1	27,990.35
Model 1	0.948	1.475	2.3	0.5	92.6	−4210.72
Model 2	0.892	1.396	1.9	0.5	93.0	−4210.18
Model 3	1.434	1.475	2.3	0.0	92.6	−4201.35
Model 4	0.942	1.503	2.3	0.5	91.9	−4198.98
Model 5	0.841	1.445	2.3	0.5	93.4	−4197.66
Model 6	0.815	1.314	1.6	1.0	94.1	−4196.15
Model 7	1.469	1.258	0.9	0.0	92.3	−4195.66
Model 8	0.952	1.202	0.7	0.0	92.3	−4191.22
Model 9	0.849	1.349	1.4	0.5	92.3	−4189.56
Model 10	0.932	1.475	2.3	0.0	92.6	−4189.56

**Table 5 vaccines-11-00263-t005:** Top 10 refined complexes of vaccine and MHC-I molecule. vdW (van der Waals energy), ACE (atomic contact energy), and HB (hydrogen bond energy).

Rank	Solution Number	Global Energy	Attractive VdW	Repulsive VdW	ACE	HB
1	1	−5.48	−5.02	0.19	1.69	0.00
2	7	2.79	−25.90	7.18	14.87	−2.19
3	9	2.98	−4.25	1.70	−2.11	0.00
4	4	7.47	−2.11	0.00	2.28	0.00
5	10	14.13	−1.94	0.00	−0.09	0.00
6	6	23.01	−42.36	98.05	10.51	−4.03
7	3	67.55	−64.86	217.80	2.60	−8.81
8	5	68.38	−38.43	155.88	3.27	−7.66
9	8	107.39	−50.90	220.11	1.40	−8.98
10	2	4497.19	−69.73	5774.65	−4.08	−10.33

**Table 6 vaccines-11-00263-t006:** Top 10 refined complexes of vaccine and MHC-II molecule. vdW (van der Waals energy), ACE (atomic contact energy), and HB (hydrogen bond energy).

Rank	Solution Number	Global Energy	Attractive VdW	Repulsive VdW	ACE	HB
1	9	0.64	−3.25	0.00	2.36	−0.27
2	2	5.11	−0.46	0.00	1.30	0.00
3	3	5.66	−0.00	0.00	0.00	0.00
4	5	16.49	−5.54	1.04	3.39	0.00
5	6	25.99	−4.47	0.00	5.30	−0.38
6	1	36.43	−4.81	1.30	4.13	0.00
7	7	68.21	−33.20	129.09	10.91	−2.55
8	10	1005.87	−47.94	1306.99	8.39	−5.44
9	4	1285.73	−45.58	1683.22	−6.34	−3.73
10	8	1663.10	−38.48	2141.13	4.25	−2.73

**Table 7 vaccines-11-00263-t007:** Top 10 refined complexes of vaccine and TLR-4 molecule. vdW (van der Waals energy), ACE (atomic contact energy), and HB (hydrogen bond energy).

Rank	Solution Number	Global Energy	Attractive VdW	Repulsive VdW	ACE	HB
1	7	−2.69	−2.65	0.00	1.51	0.00
2	3	0.53	−25.46	10.39	10.38	−1.62
3	9	1.05	−6.07	2.39	−1.62	−0.33
4	2	6.64	−39.02	35.23	16.74	−7.23
5	8	24.74	−12.29	8.51	7.35	−1.00
6	6	34.77	−18.81	7.52	17.81	−0.76
7	5	113.53	−41.79	199.10	8.01	−5.16
8	4	355.02	−21.74	471.02	−3.09	−1.73
9	10	452.38	−29.95	571.85	11.56	−4.40
10	1	4146.04	−63.37	5273.51	10.33	−11.76

**Table 8 vaccines-11-00263-t008:** Binding free energies calculation. The units are kcal/mol.

Energy Parameter	TLR-4-Vaccine Complex	MHC-I-Vaccine Complex	MHC-II-Vaccine Complex
MM-GBSA			
VDWAALS	−150.96	−137.99	−131.57
EEL	−96.37	−81.61	−80.22
EGB	35.00	32.08	18.00
ESURF	−22.85	−19.27	−21.94
Delta G gas	−247.33	−219.6	−211.79
Delta G solv	12.15	12.81	−3.94
Delta Total	−259.48	−206.79	−215.73
MM-PBSA			
VDWAALS	−150.96	−137.99	−131.57
EEL	−96.37	−81.61	−80.22
EPB	35.00	32.08	18.00
ENPOLAR	−22.85	−19.27	−21.94
Delta G gas	−247.33	−219.6	−211.79
Delta G solv	12.15	12.81	−3.94
Delta Total	−235.18	−206.79	−215.73

## Data Availability

All the data is available in this study.

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
