# Peer review of "Proteome-Wide Screening of Potential Vaccine Targets against Brucella melitensis"

_vaccines, 2023, doi:10.3390/vaccines11020263_

Round 1
Reviewer 1 Report (Previous Reviewer 2)
Journal Vaccines (ISSN 2076-393X)
Manuscript ID vaccines-2084278
Type Article
Title Proteome Wide Screening of Potential Vaccine Targets against Brucella melitensis
The article has been already submitted for revision and the following comments refer to revised version submitted.
In general, the article has been improved but there are section of introduction and discussion that need to be modified.
Specific comments
Abstract
Line 24: I would add these sentences after “yaks and dogs”: “Infected animals also represent the major source of infection for humans. Development of safer and effective vaccines for brucellosis remain a priority to support disease control and eradication in animals and to prevent infection to humans” (Add references/check for reviews)
Introduction
In the introduction, it is necessary to give a minimal background on brucellosis and on the current gap/gaps authors want to cover with their research and that would support the work done. Some of the information on brucellosis included in the abstract are lacking in the introduction. Example: “Brucella melitensis is an etiological agent of brucellosis, which mostly affects sheep’s and goats but several cases have also been reported in cattle, water buffalo, yaks and dogs”. This information is lacking in the introduction.
Line 75-87. Sentences for this part have to be modified. See suggestion below
Line 77-78. “The pathogen is responsible several diseases including but not limited to brucellosis, and malta fever”. Meaning of this sentence is not clear. Reformulate.
Line 80. I would suggest to add a sentence like this: Infected animals represent the major source of infection for humans, through direct exposure or due to consumption of contaminated and unpasteurised dairy products. In endemic areas vaccination of susceptible animals would reduce disease prevalence also limiting the risk of disease transmission to humans (references).
Line 83. I would add this sentence “Rev1 is a live attenuated vaccines and despite its efficacy, several drawbacks remain due to its residual pathogenicity”.
Line 84. I would start the sentence as follows: “Previous attempts to produce safer and effective recombinant brucella vaccines was described by xx et al that included two antigens (periplasmic bp26 and chaperone trigger factor proteins)” Despite of these efforts……
Material and methods and Results
No comments
Discussion
The discussion section still require major revision considering the following points:
Line 433-439. This are examples of application of reverse vaccinology but then the authors should discuss their findings focusing on brucellosis or stating that there are no other studies to compare with. The authors are describing a prototype vaccine for Brucellosis and the readers are expecting to have some comparison with other studies that focus on Brucella vaccines. Thus, the authors should discuss the selected molecules composing the multiepitope vaccine against B. melitensis. This part is still missing. The readers should know if the molecules selected have already been described as immunogenic (in humans or animals) or tested in previous vaccine formulations. In addition, it would be interesting to discuss if the designed prototype vaccine would potentially protect against infections by other zoonotic smooth brucella like B. abortus or B. suis, considering the high homology conserved among brucella species, suggesting the broad use of the vaccine to control disease caused by different brucella strains. (The authors can easily check if the selected molecules are also present in other brucella species).
Another point of interest to be discussed is how the immunization with this multiepitope vaccine would influence results of current diagnostic tests used for diagnosis of classical brucellosis (caused by smooth brucella species like B. melitensis and B. abortus). Would it be possible to distinguish infected from vaccinated patients? (Most of the serological tests for brucellosis uses the LPS component as antigen and have been developed to detect antibodies elicited against the LPS of smooth brucella during infection such as B. melitensis and B. abortus. Because the prototype vaccine does not include the LPS component the authors may suppose that vaccinated animals will not produce antibodies against the LPS component and should test negative to the classical serological tests such as the Rose Bengal test and the complement fixation test. Some information on this topis can also found in the Manual for diagnostic tests and Vaccines of the World Organisation for Animal Health, former OIE, chapter 3.1.4). This may support the application of the proposed prototype vaccine for a DIVA (Differentiating Infected from Vaccinated Animals) approach and that would be ideal for disease control.
More information on the nature and application of adjiuvant (cholera toxin-b) should be provided to support the choice of this molecule. What is advantage of using the cholera toxin-b instead of other adjuvants? What type of immunity is expected to elicit considering that protection to brucella infection requires a cell mediated immunity?
Conclusions
Line 448-457. This part is a summary of results and should be moved at the beginning pf the discussion section. In the conclusions the authors should focus on how the new knowledge may be applied and suggests future directions of research and development. More specifically, conclusions should be focused on the added value of the approach considered and provide indication on how reverse vaccinology can be improved to become a more realistic model for vaccine development. The in silico modelling of a prototype vaccine should include all physical, chemical and biological elements that may influence the vaccine efficacy and the biological response induced.
Author Response
Response to Reviewer Comments We thank the Referee for spending time and interest in our work and for helpful comments that will greatly improve the manuscript. We have checked all the general and specific comments provided by the Referee and have made all the necessary changes according to his indications. Please refer to yellow highlighted sections in the revised manuscript. Reviewer 1 Comments and Suggestions for Authors Journal Vaccines (ISSN 2076-393X) Manuscript ID vaccines-2084278 Type Article Title Proteome Wide Screening of Potential Vaccine Targets against Brucella melitensis The article has been already submitted for revision and the following comments refer to revised version submitted. In general, the article has been improved but there are section of introduction and discussion that need to be modified. Response: The introduction and discussion sections have been revised as per reviewer comments. Specific comments Abstract Line 24: I would add these sentences after “yaks and dogs”: “Infected animals also represent the major source of infection for humans. Development of safer and effective vaccines for brucellosis remain a priority to support disease control and eradication in animals and to prevent infection to humans” (Add references/check for reviews). Response: Thank you for the valuable suggestion. The sentence has been added to the revised manuscript abstract section. Introduction. In the introduction, it is necessary to give a minimal background on brucellosis and on the current gap/gaps authors want to cover with their research and that would support the work done. Some of the information on brucellosis included in the abstract are lacking in the introduction. Example: “Brucella melitensis is an etiological agent of brucellosis, which mostly affects sheep’s and goats but several cases have also been reported in cattle, water buffalo, yaks and dogs”. This information is lacking in the introduction. Response: Done in the revised manuscript. Line 75-87. Sentences for this part have to be modified. See suggestion below. Response: Done in the revised manuscript as per reviewer suggestion. Line 77-78. “The pathogen is responsible several diseases including but not limited to brucellosis, and malta fever”. Meaning of this sentence is not clear. Reformulate. Response: Reformatting of the sentence is done in the modified version. Line 80. I would suggest to add a sentence like this: Infected animals represent the major source of infection for humans, through direct exposure or due to consumption of contaminated and unpasteurised dairy products. In endemic areas vaccination of susceptible animals would reduce disease prevalence also limiting the risk of disease transmission to humans (references). Response: Done in the revised manuscript. Line 83. I would add this sentence “Rev1 is a live attenuated vaccines and despite its efficacy, several drawbacks remain due to its residual pathogenicity”. Response: Done in the revised manuscript. Line 84. I would start the sentence as follows: “Previous attempts to produce safer and effective recombinant brucella vaccines was described by xx et al that included two antigens (periplasmic bp26 and chaperone trigger factor proteins)” Despite of these efforts…… Response: Done in the revised manuscript. Material and methods and Results No comments Discussion The discussion section still require major revision considering the following points: Line 433-439. This are examples of application of reverse vaccinology but then the authors should discuss their findings focusing on brucellosis or stating that there are no other studies to compare with. The authors are describing a prototype vaccine for Brucellosis and the readers are expecting to have some comparison with other studies that focus on Brucella vaccines. Thus, the authors should discuss the selected molecules composing the multiepitope vaccine against B. melitensis. This part is still missing. The readers should know if the molecules selected have already been described as immunogenic (in humans or animals) or tested in previous vaccine formulations. In addition, it would be interesting to discuss if the designed prototype vaccine would potentially protect against infections by other zoonotic smooth brucella like B. abortus or B. suis, considering the high homology conserved among brucella species, suggesting the broad use of the vaccine to control disease caused by different brucella strains. (The authors can easily check if the selected molecules are also present in other brucella species). Response: Thank you for the comment. The previous bioinformatic studies on B. melitensis are added to revised manuscript discussion section. The study was designed to mainly target B. melitensis strains therefore other species are not objective of the study. Another point of interest to be discussed is how the immunization with this multiepitope vaccine would influence results of current diagnostic tests used for diagnosis of classical brucellosis (caused by smooth brucella species like B. melitensis and B. abortus). Would it be possible to distinguish infected from vaccinated patients? (Most of the serological tests for brucellosis uses the LPS component as antigen and have been developed to detect antibodies elicited against the LPS of smooth brucella during infection such as B. melitensis and B. abortus. Because the prototype vaccine does not include the LPS component the authors may suppose that vaccinated animals will not produce antibodies against the LPS component and should test negative to the classical serological tests such as the Rose Bengal test and the complement fixation test. Some information on this topis can also found in the Manual for diagnostic tests and Vaccines of the World Organisation for Animal Health, former OIE, chapter 3.1.4). This may support the application of the proposed prototype vaccine for a DIVA (Differentiating Infected from Vaccinated Animals) approach and that would be ideal for disease control. Response: Thank you for the useful comment. However, the authors are unable to understand the point and think that this might not be useful for discussion in this paper. More information on the nature and application of adjiuvant (cholera toxin-b) should be provided to support the choice of this molecule. What is advantage of using the cholera toxin-b instead of other adjuvants? What type of immunity is expected to elicit considering that protection to brucella infection requires a cell mediated immunity? Response: Done in the revised manuscript section 3.4.Reviewer 2 Report (New Reviewer)
The manuscript describes a detailed approach to design a subunit vaccine for brucellosis using computational tools. It is great if the following points are addressed in the ‘introduction’ and ‘discussion’ sections…
· - What are the drawbacks of the already made/used vaccines such as Rev1 and bp6? Why you need a new vaccine?
· - What are the anticipated advantages of your designed vaccine over Rev1 and bp6 (enhanced protection, enhanced safety, etc.?)
· - Any suggestions of possible route(s) of administration of your designed vaccine?
· - What could be the potential drawbacks of your vaccine?
The manuscript carries many editorial and typing errors. It needs to be carefully reviewed and edited prior to publication. A few of the needed corrections are shown below.
· Line 22 design
· Line 23 sheep
· Line 82 goat
· Line 83 elicit
· Line 162 attach
· Line 191 Peptide vaccines having weak ability to provoke host immune response (incomplete sentence)
· Line 192 handled
· Line 409 values
· Lines 419 and 420 losing their efficacy and development of new is of less attention due to due to several challenging regulatory challenges faced by pharma companies (incomplete sentence)
· Line 430 should be 500,000
Author Response
Reviewer 2
Comments and Suggestions for Authors
The manuscript describes a detailed approach to design a subunit vaccine for brucellosis using computational tools. It is great if the following points are addressed in the ‘introduction’ and ‘discussion’ sections…
- - What are the drawbacks of the already made/used vaccines such as Rev1 and bp6? Why you need a new vaccine?
Response: Done in the revised manuscript introduction section.
- - What are the anticipated advantages of your designed vaccine over Rev1 and bp6 (enhanced protection, enhanced safety, etc.?).
Response: Done in the revised manuscript discussion section.
Any suggestions of possible route(s) of administration of your designed vaccine?
Response: Done in the revised manuscript discussion section.
- - What could be the potential drawbacks of your vaccine?
Response: Done in the revised manuscript discussion section.
The manuscript carries many editorial and typing errors. It needs to be carefully reviewed and edited prior to publication. A few of the needed corrections are shown below.
Response: The manuscript is thoroughly revised for English language and grammatical errors.
- Line 22 design
Response: Corrected in the revised manuscript.
- Line 23 sheep
Response: Corrected in the revised manuscript.
- Line 82 goat
Response: Corrected in the revised manuscript.
- Line 83 elicit
Response: Corrected in the revised manuscript.
- Line 162 attach
Response: Corrected in the revised manuscript.
- Line 191 Peptide vaccines having weak ability to provoke host immune response (incomplete sentence).
Response: Corrected in the revised manuscript.
- Line 192 handled
Response: Corrected in the revised manuscript.
- Line 409 values
Response: Corrected in the revised manuscript.
- Lines 419 and 420 losing their efficacy and development of new is of less attention due to due to several challenging regulatory challenges faced by pharma companies (incomplete sentence)
Response: Corrected in the revised manuscript.
- Line 430 should be 500,000
Response: Corrected in the revised manuscript.
Round 2
Reviewer 1 Report (Previous Reviewer 2)
The paper has been imporved and additional information provided in the different sections increased the readibility of the article.
Minor comments
Line 266 change "to reconfirmed" in "to reconfirm"
Line 321. A preposition is missing in the sentence. ...which is present (on?) antigen presenting cells......
Line 422. Change ";" with ":" Line 442-443. Sentence with no clear meaning please re-phrase Line 432. Change "recognitiona" in "recognition"
Line 508. Change "valid" with "validate" or with "assess" Line 517. Check the verb tense Line 522-523. Please rephrase to explain better. Also avoid repetition of the word "simple" in the sentence, look for synonyms
Line 522. Correct "to designed" in "to design"
Line 525. format as italics "in vivo" and "in vitro"
Author Response
Response to Reviewer Comments
We thank the Referee for spending time and interest in our work and for helpful comments that will greatly improve the manuscript. We have checked all the general and specific comments provided by the Referee and have made all the necessary changes according to his indications. Please refer to green highlighted sections in the revised manuscript.
Comments and Suggestions for Authors
The paper has been imporved and additional information provided in the different sections increased the readibility of the article.
Minor comments
Line 266 change "to reconfirmed" in "to reconfirm"
Response: Done in revised manuscript.
Line 321. A preposition is missing in the sentence. ...which is present (on?) antigen presenting cells......
Response: Done in revised manuscript.
Line 422. Change ";" with ":"
Response: Done in revised manuscript.
Line 442-443. Sentence with no clear meaning please re-phrase.
Response: Done in revised manuscript.
Line 432. Change "recognitiona" in "recognition"
Response: Done in revised manuscript.
Line 508. Change "valid" with "validate" or with "assess"
Response: Done in revised manuscript.
Line 517. Check the verb tense Line 522-523. Please rephrase to explain better. Also avoid repetition of the word "simple" in the sentence, look for synonyms
Response: Done in revised manuscript.
Line 522. Correct "to designed" in "to design"
Response: Done in revised manuscript.
Line 525. format as italics "in vivo" and "in vitro"
Response: Done in revised manuscript.
This manuscript is a resubmission of an earlier submission. The following is a list of the peer review reports and author responses from that submission.
Round 1
Reviewer 1 Report
The manuscript entitled "Proteome Wide Screening of Potential Vaccine Targets against 2 Brucella melitensis" attempts to design a multi-epitope vaccine against B. melitensis using in silico and immunoinformatic approaches. Identifying antigens and creating vaccine immunogens using in silico approaches without evaluating their in vitro expression and immunogenicity is not recommended. Work done in the manuscript is not enough of an advance for the journal and lacks experimental data to support the hypotheses. Since the concept of the manuscript is identical to the authors' previously published studies (Ahmad et al., 2021, Vaccines, 10.3390/vaccines9030293; Ismail et al., 2020, Eur J Pharm Sci, DOI: 10.1016/j.ejps.2020.105258; Ahmad et al., 2019, Eur J Pharm Sci, DOI: 10.1016/j.ejps.2019.02.023; Ahmad et al., 2018, J Mol Graph Model, DOI: 10.1016/j.jmgm.2018.04.020; Dar et al., 2021, Sci Rep, DOI: 10.1038/s41598-021-90868-2; Asad et al., 2018, J Mol Graph Model, DOI: 10.1016/j.jmgm.2018.01.010 ), the work presented in the manuscript has no originality and novelty. The manuscript is poorly written and lacks experimental validation and the proper graphical representation of the data. It is difficult to follow the rationale of the manuscript. A grammatical revision will benefit the text.
The Introduction section doesn't include sufficient information about the manuscript's purpose and importance.
Additionally, the construct shown in figures 5 and 6 does not make sense. Authors incorporated 17 disulfide bonds in a structure to maintain structural stability. Proteins with disulfide bonds need to be refolded correctly from inclusion bodies when expressed in E. coli. It is practically impossible to express/correctly refold such a construct.
There are no units for binding free energy values throughout the manuscript.
Author Response
Response to Reviewer Comments
We thank the Referee for spending time and interest in our work and for helpful comments that will greatly improve the manuscript. We have checked all the general and specific comments provided by the Referee and have made all the necessary changes according to his indications. Please refer to yellow highlighted sections in the revised manuscript.
Reviewer # 1
Comments and Suggestions for Authors
The manuscript entitled "Proteome Wide Screening of Potential Vaccine Targets against 2 Brucella melitensis" attempts to design a multi-epitope vaccine against B. melitensis using in silico and immunoinformatic approaches. Identifying antigens and creating vaccine immunogens using in silico approaches without evaluating their in vitro expression and immunogenicity is not recommended. Work done in the manuscript is not enough of an advance for the journal and lacks experimental data to support the hypotheses. Since the concept of the manuscript is identical to the authors' previously published studies (Ahmad et al., 2021, Vaccines, 10.3390/vaccines9030293; Ismail et al., 2020, Eur J Pharm Sci, DOI: 10.1016/j.ejps.2020.105258; Ahmad et al., 2019, Eur J Pharm Sci, DOI: 10.1016/j.ejps.2019.02.023; Ahmad et al., 2018, J Mol Graph Model, DOI: 10.1016/j.jmgm.2018.04.020; Dar et al., 2021, Sci Rep, DOI: 10.1038/s41598-021-90868-2; Asad et al., 2018, J Mol Graph Model, DOI: 10.1016/j.jmgm.2018.01.010 ), the work presented in the manuscript has no originality and novelty. The manuscript is poorly written and lacks experimental validation and the proper graphical representation of the data. It is difficult to follow the rationale of the manuscript. A grammatical revision will benefit the text.
Response: Thank you for the valuable comment. The manuscript is revised in light of reviewer comments. We understand the reviewer concern. As the scope of the paper is computational, the main objective of the work is to provide theoretical vaccine model for experimentalists to check the designed vaccine immune protective efficacy against the said pathogen in vivo. Reverse Vaccinology (RV), has received more attention in recent years and has been used for the identification of vaccine proteins against different pathogens [1]. The RV approach was first applied to the bacterial pathogen Meningococcus B (MenB) and led to the license Bexsero vaccine [2], where RV played a significant role in screening for an antigen with the broadest bactericidal activity and ultimately resolved the long journey of MenB vaccine development. RV has also been applied to many other bacterial pathogens, including group A Streptococcus, antibiotic-resistant Staphylococcus aureus, Streptococcus pneumonia, and Chlamydia. The efficacy of peptide or subunit based vaccines initially identified through a RV protocol has also been proven experimentall [3,4]. In this study, a RV approach was used to screen possible vaccine proteins against the Brucella melitensis, identifying. Nine proteins were identified as strong candidates for vaccine development. Experimental follow up by testing the immune protection efficacy of the screened epitopes in animal models will open for experimentalists and this study will definitely speed up vaccine development process against this pathogen. This text has been highlighted in the introduction section of the revised manuscript.
References
- Ong E, Wong MU, Huffman A, He Y. COVID-19 coronavirus vaccine design using reverse vaccinology and machine learning. bioRxiv [Preprint]. 2020 Mar 21:2020.03.20.000141. doi: 10.1101/2020.03.20.000141. Update in: Front Immunol. 2020 Jul 03;11:1581. PMID: 32511333; PMCID: PMC7239068.
- Folaranmi T, Rubin L, Martin SW, Patel M, MacNeil JR; Centers for Disease Control (CDC). Use of Serogroup B Meningococcal Vaccines in Persons Aged ≥10 Years at Increased Risk for Serogroup B Meningococcal Disease: Recommendations of the Advisory Committee on Immunization Practices, 2015. MMWR Morb Mortal Wkly Rep. 2015 Jun 12;64(22):608-12. Erratum in: MMWR Morb Mortal Wkly Rep. 2015 Jul 31;64(29):806. PMID: 26068564; PMCID: PMC4584923.
- Maione D, Margarit I, Rinaudo CD, Masignani V, Mora M, Scarselli M, Tettelin H, Brettoni C, Iacobini ET, Rosini R, D'Agostino N, Miorin L, Buccato S, Mariani M, Galli G, Nogarotto R, Nardi-Dei V, Vegni F, Fraser C, Mancuso G, Teti G, Madoff LC, Paoletti LC, Rappuoli R, Kasper DL, Telford JL, Grandi G. Identification of a universal Group B streptococcus vaccine by multiple genome screen. Science. 2005 Jul 1;309(5731):148-50. doi: 10.1126/science.1109869. Erratum in: Science. 2013 Jan 11;339(6116):141. Nardi Dei, Vincenzo [corrected to Nardi-Dei, Vincenzo]. PMID: 15994562; PMCID: PMC1351092.
- Sette A, Rappuoli R. Reverse vaccinology: developing vaccines in the era of genomics. Immunity. 2010 Oct 29;33(4):530-41. doi: 10.1016/j.immuni.2010.09.017. PMID: 21029963; PMCID: PMC3320742.
Reviewer 2 Report
Manuscript ID vaccines-1922458
Title Proteome Wide Screening of Potential Vaccine Targets against Brucella melitensis
The article describes all the steps of in-silico design of a multi-epitopes vaccine for B. melitensis, covering from retrieval of B. melitensis sequences, to functional simulation of the designed vaccine with immune receptors. The methodology applied consider several aspects of vaccine development, not limited to antigen selection but also the final construction of the vaccine molecule, providing all the necessary elements for production of a vaccine prototype. Development of a vaccine B. melitensis would contribute to counteract infection in humans also limiting the risks of antimicrobial resistance.
The article presents a very interesting pipeline for development of a multiepitope vaccine to prevent brucella infection due to B. melitensis. As stated in the introduction, brucellosis caused by classical (smooth) brucella species (B. melitensis and B. abortus) is primarily an animal disease but also a zoonosis and majority of acquired infection in humans are due to direct exposure to infectious material or infected animals or due to consumption or infected raw milk or unpasteurised dairy products. Transmission from human to human is rare to occur, with few exceptions. This means that especially in areas where animal brucellosis is endemic, to prevent infection in humans, it is crucial to control or prevent brucella infection in animals. To date, only live vaccines are available for animals with several drawbacks and in terms of vaccine development a new generation brucella vaccine for animals would a greater impact than a vaccine for humans, at same time reducing the risk of infection for humans.
In the introduction and in the aim of the study it is not clearly stated if the multiepitope vaccine designed has been developed exclusively for humans or with potential applications in animals. The reverse vaccinology approach presented in the paper may represent a guide for brucella vaccine development. However, in order to have a greater and broader impact on scientific community working on brucella, the article should also cover and discuss possible application or limitations of this approach on target animal species, mainly sheep, goats and cattle, tha major source of brucella infection for humans. In this respect, one of the aspects to be covered or at least discussed is if the pipeline developed to design a vaccine prototype can be applied directly or easily to these animal species.
The discussion is not focus on study results and most of the text provide methodology information, not considering several relevant aspects that should be considered and discussed (see detailed comment for the discussion section)
Specific comments
Introduction
Major comments
The introduction should include a paragraph stating which are the challenges in Brucella vaccine development that can be overcome by reverse vaccinology
It is important to clarify if the study focused only on vaccine targets for human only or potentially for both humans and animals
Minor comments
Line 74-77 check past participle of the verbs
Material and methods
Very well structured and easy to read. Figure 1 is very helpful to clarify the entire methodology applied.
Minor comments
Line 187 the verb is missing
Line 189-191: References are missing for the Virulentpred and ToxiPred methods
Information on RMSD and RMSF analysis are missing in M&M section but presented in the results in the Molecular dynamic simulation section Lines 371, 372 and 383)
Results
Well structured section
Line 323 check grammar (reverted)
Discussion
As mentioned above discussion section should be revised considering the following points:
Provide information on other studies that attempted to develop human vaccines and possibly indicate why so far no vaccines have been developed for humans
Discuss on selected molecules composing the multiepitope vaccine against B. melitensis. This part is missing. The readers should know is these molecules have already been described as immunogenic (in humans or also animals) or tested in previous vaccine formulations. In addition, it would be interesting to discuss if the designed prototype vaccine would potentially protect against infections by other zoonotic smooth brucella like B. abortus or B. suis, considering the high homology conserved among brucella species.
Another point of interest to be discussed is how the immunization with this multiepitope vaccine would influence results of current diagnostic tests used for diagnosis of classical brucellosis (caused by smooth brucella species like B. melitensis and B. abortus). Would it be possible to distinguish infected from vaccinated patients?
More information on the nature and application of adjiuvant (cholera toxin-b) should be provided to support the choice of this molecule. What is advantage of using the cholera toxin-b instead of other adjuvants? What type of immunity does it promote? Can be used to differentiate vaccination from infection?
What are the major limitations/challenges of this approach that have to be considered when moving from in silico to a prototype product for in vitro/in vivo testing?
Any example of in silico designed vaccines (using this approach) that turned to a real commercial vaccine?
Specific comments
Line 401 Pasture vaccinology method… (meaning Pasteur?)
Line 408-431 describe methodology and should be replaced with results discussion
Conclusions
Conclusions should be focused on the added value of the approach considered and provide indication on how reverse vaccinology can be improved to become e more realistic model for vaccine development. The in silico modelling of a prototype vaccine should include all physical, chemical and biological elements that may influence
Author Response
Response to Reviewer Comments
We thank the Referee for spending time and interest in our work and for helpful comments that will greatly improve the manuscript. We have checked all the general and specific comments provided by the Referee and have made all the necessary changes according to his indications. Please refer to yellow highlighted sections in the revised manuscript.
Reviewer # 2
Comments and Suggestions for Authors
Manuscript ID vaccines-1922458
Title Proteome Wide Screening of Potential Vaccine Targets against Brucella melitensis
The article describes all the steps of in-silico design of a multi-epitopes vaccine for B. melitensis, covering from retrieval of B. melitensis sequences, to functional simulation of the designed vaccine with immune receptors. The methodology applied consider several aspects of vaccine development, not limited to antigen selection but also the final construction of the vaccine molecule, providing all the necessary elements for production of a vaccine prototype. Development of a vaccine B. melitensis would contribute to counteract infection in humans also limiting the risks of antimicrobial resistance.
The article presents a very interesting pipeline for development of a multiepitope vaccine to prevent brucella infection due to B. melitensis. As stated in the introduction, brucellosis caused by classical (smooth) brucella species (B. melitensis and B. abortus) is primarily an animal disease but also a zoonosis and majority of acquired infection in humans are due to direct exposure to infectious material or infected animals or due to consumption or infected raw milk or unpasteurised dairy products. Transmission from human to human is rare to occur, with few exceptions. This means that especially in areas where animal brucellosis is endemic, to prevent infection in humans, it is crucial to control or prevent brucella infection in animals. To date, only live vaccines are available for animals with several drawbacks and in terms of vaccine development a new generation brucella vaccine for animals would a greater impact than a vaccine for humans, at same time reducing the risk of infection for humans.
Response: All information is provided in the revised manuscript.
In the introduction and in the aim of the study it is not clearly stated if the multiepitope vaccine designed has been developed exclusively for humans or with potential applications in animals. The reverse vaccinology approach presented in the paper may represent a guide for brucella vaccine development. However, in order to have a greater and broader impact on scientific community working on brucella, the article should also cover and discuss possible application or limitations of this approach on target animal species, mainly sheep, goats and cattle, tha major source of brucella infection for humans. In this respect, one of the aspects to be covered or at least discussed is if the pipeline developed to design a vaccine prototype can be applied directly or easily to these animal species.
Response: All information is provided in the revised manuscript.
The discussion is not focus on study results and most of the text provide methodology information, not considering several relevant aspects that should be considered and discussed (see detailed comment for the discussion section)
Response: All information is provided in the revised manuscript.
Specific comments
Introduction
Major comments
The introduction should include a paragraph stating which are the challenges in Brucella vaccine development that can be overcome by reverse vaccinology
Response: The information is provided in the revised manuscript.
It is important to clarify if the study focused only on vaccine targets for human only or potentially for both humans and animals
Response: The information is provided in the revised manuscript.
Minor comments
Line 74-77 check past participle of the verbs
Response: Corrections are done in the revised manuscript.
Material and methods
Very well structured and easy to read. Figure 1 is very helpful to clarify the entire methodology applied.
Minor comments
Line 187 the verb is missing
Response: Corrections are done in the revised manuscript.
Line 189-191: References are missing for the Virulentpred and ToxiPred methods
Response: Web link is added to each tool in the revised manuscript.
Information on RMSD and RMSF analysis are missing in M&M section but presented in the results in the Molecular dynamic simulation section Lines 371, 372 and 383)
Response: Added to the revised manuscript section 2.24.
Results
Well structured section
Line 323 check grammar (reverted)
Response: Corrected in the revised manuscript.
Discussion
As mentioned above discussion section should be revised considering the following points:
Provide information on other studies that attempted to develop human vaccines and possibly indicate why so far no vaccines have been developed for humans
Discuss on selected molecules composing the multiepitope vaccine against B. melitensis. This part is missing. The readers should know is these molecules have already been described as immunogenic (in humans or also animals) or tested in previous vaccine formulations. In addition, it would be interesting to discuss if the designed prototype vaccine would potentially protect against infections by other zoonotic smooth brucella like B. abortus or B. suis, considering the high homology conserved among brucella species.
Another point of interest to be discussed is how the immunization with this multiepitope vaccine would influence results of current diagnostic tests used for diagnosis of classical brucellosis (caused by smooth brucella species like B. melitensis and B. abortus). Would it be possible to distinguish infected from vaccinated patients?
More information on the nature and application of adjiuvant (cholera toxin-b) should be provided to support the choice of this molecule. What is advantage of using the cholera toxin-b instead of other adjuvants? What type of immunity does it promote? Can be used to differentiate vaccination from infection?
Response: All information is provided in the revised manuscript.
What are the major limitations/challenges of this approach that have to be considered when moving from in silico to a prototype product for in vitro/in vivo testing?
Any example of in silico designed vaccines (using this approach) that turned to a real commercial vaccine?
Response: Already mentioned in the introduction section.
Specific comments
Line 401 Pasture vaccinology method… (meaning Pasteur?)
Response: Corrected in the revised manuscript.
Line 408-431 describe methodology and should be replaced with results discussion
Response: The mentioned text is removed and proper results/discussion is added.
Conclusions
Conclusions should be focused on the added value of the approach considered and provide indication on how reverse vaccinology can be improved to become e more realistic model for vaccine development. The in silico modelling of a prototype vaccine should include all physical, chemical and biological elements that may influence
Response: The conclusion section is improved in light of the revised manuscript.
Round 2
Reviewer 1 Report
Thanks to the authors for their responses and additional modifications to the manuscript. Again, the work done in the manuscript is not enough of an advance for the journal, and there are no experimental data to support the hypotheses. Validating the results of computational work is of the utmost importance. It is at least necessary to include expression, purification, and preliminary characterization of the designed vaccine construct. I agree with the author's statements regarding the importance of Reverse Vaccinology. The same approaches have been described in a number of papers for different pathogens. It is practically impossible for experimentalists to investigate such hundreds of thousands of designed constructs. Therefore, it is imperative to validate these constructs for their expression and purification so that vaccine development can be accelerated. When such designed constructs cannot be expressed and purified, a study is of no use. Accordingly, I will continue to respond in the same way as before. Since the concept of the manuscript is identical to the authors' recently published studies (Ahmad et al., 2021, Vaccines, 10.3390/vaccines9030293; Ismail et al., 2020, Eur J Pharm Sci, DOI: 10.1016/j.ejps.2020.105258; Ahmad et al., 2019, Eur J Pharm Sci, DOI: 10.1016/j.ejps.2019.02.023; Ahmad et al., 2018, J Mol Graph Model, DOI: 10.1016/j.jmgm.2018.04.020; Dar et al., 2021, Sci Rep, DOI: 10.1038/s41598-021-90868-2; Asad et al., 2018, J Mol Graph Model, DOI: 10.1016/j.jmgm.2018.01.010 ), the work presented in the manuscript lacks originality and novelty. This manuscript will benefit from experimental validation of the in silico study described.
Author Response
Response to Reviewer Comments
We thank the Referee for spending time and interest in our work and for helpful comments that will greatly improve the manuscript. We have checked all the general and specific comments provided by the Referee and have made all the necessary changes according to his indications.
Response: Thank you for the valuable comment. We again understand the reviewer concern. As the scope of the paper is computational, the main objective of the work is to provide theoretical vaccine model for experimentalists to check the designed vaccine immune protective efficacy against the said pathogen in vivo. The rationale behind submitting this paper to "MDPI Vaccine " is due to the fact that journal scope also consider pure computational works. Yes, experimental validation is needed but that is beyond the goal of this study and we are a pure computational research team and experiments are not our within domain. We think the findings will speed up vaccine development against the pathogen and will experimentalists.
Reverse Vaccinology (RV), has received more attention in recent years and has been used for the identification of vaccine proteins against different pathogens [1]. The RV approach was first applied to the bacterial pathogen Meningococcus B (MenB) and led to the license Bexsero vaccine [2], where RV played a significant role in screening for an antigen with the broadest bactericidal activity and ultimately resolved the long journey of MenB vaccine development. RV has also been applied to many other bacterial pathogens, including group A Streptococcus, antibiotic-resistant Staphylococcus aureus, Streptococcus pneumonia, and Chlamydia. The efficacy of peptide or subunit based vaccines initially identified through a RV protocol has also been proven experimentall [3,4]. In this study, a RV approach was used to screen possible vaccine proteins against the Brucella melitensis, identifying. Nine proteins were identified as strong candidates for vaccine development. Experimental follow up by testing the immune protection efficacy of the screened epitopes in animal models will open for experimentalists and this study will definitely speed up vaccine development process against this pathogen.
References
- Ong E, Wong MU, Huffman A, He Y. COVID-19 coronavirus vaccine design using reverse vaccinology and machine learning. bioRxiv [Preprint]. 2020 Mar 21:2020.03.20.000141. doi: 10.1101/2020.03.20.000141. Update in: Front Immunol. 2020 Jul 03;11:1581. PMID: 32511333; PMCID: PMC7239068.
- Folaranmi T, Rubin L, Martin SW, Patel M, MacNeil JR; Centers for Disease Control (CDC). Use of Serogroup B Meningococcal Vaccines in Persons Aged ≥10 Years at Increased Risk for Serogroup B Meningococcal Disease: Recommendations of the Advisory Committee on Immunization Practices, 2015. MMWR Morb Mortal Wkly Rep. 2015 Jun 12;64(22):608-12. Erratum in: MMWR Morb Mortal Wkly Rep. 2015 Jul 31;64(29):806. PMID: 26068564; PMCID: PMC4584923.
- Maione D, Margarit I, Rinaudo CD, Masignani V, Mora M, Scarselli M, Tettelin H, Brettoni C, Iacobini ET, Rosini R, D'Agostino N, Miorin L, Buccato S, Mariani M, Galli G, Nogarotto R, Nardi-Dei V, Vegni F, Fraser C, Mancuso G, Teti G, Madoff LC, Paoletti LC, Rappuoli R, Kasper DL, Telford JL, Grandi G. Identification of a universal Group B streptococcus vaccine by multiple genome screen. Science. 2005 Jul 1;309(5731):148-50. doi: 10.1126/science.1109869. Erratum in: Science. 2013 Jan 11;339(6116):141. Nardi Dei, Vincenzo [corrected to Nardi-Dei, Vincenzo]. PMID: 15994562; PMCID: PMC1351092.
- Sette A, Rappuoli R. Reverse vaccinology: developing vaccines in the era of genomics. Immunity. 2010 Oct 29;33(4):530-41. doi: 10.1016/j.immuni.2010.09.017. PMID: 21029963; PMCID: PMC3320742.